# Home Away from Home: Comparing Factors Impacting Migrants’ and Italians Sense of Psychological Home

**DOI:** 10.3390/bs12100387

**Published:** 2022-10-10

**Authors:** Andrew P. Camilleri, Joseph R. Ferrari, Vittoria Romoli, Paola Cardinali, Laura Migliorini

**Affiliations:** 1Department of Psychology, DePaul University, Chicago, IL 60614, USA; 2Department of Education Sciences, University of Genoa, 16121 Genoa, Italy; 3Department of Economics, Universitas Mercatorum, 00186 Rome, Italy

**Keywords:** migration, psychological home

## Abstract

Psychological home is an understudied concept within community psychology, especially focused on migrants. Previous literature on psychological home found a positive relationship with well-being and resilience in general populations and migrants. Studying psychological home in migrants may provide important tools to buffer various stresses associated with migration. The present study explored the relationship between psychological home and demographic factors, including dwelling type and situation between migrants (*n* = 132) and Italian citizens’ population (*n* = 76). Results offer theoretical reasons explaining the differences in the meaning of home between migrant and non-immigrant populations.

## 1. Introduction

The term home evokes multiple meanings when used in everyday language. Vand der Klis and Karsten [1] identify three frameworks to understand the concept of home, namely the physical dwelling itself, activities surrounding the dwelling, and particular social settings that are familiar to the person around their dwelling area. Literature from a variety of fields indicated that the concept of home frequently was associated with feelings of safety and security within the individuals that partake within the concept [2] however when the experience of home is burdened by over-abundance of clutter, this feeling may result in a lower sense of well-being [3]. Such associations highlight the importance of the home concept as a construct worth studying for community psychologists who are predominantly interested in how living environments impacts the well-being of individuals and communities [4].

The concept of home takes on particular importance in the context of migration, defined as the movement of people around the world, and may be due to push/pull factors within the context or origin and the context of destination [5]. Migrants frequently experience significant acculturative stresses as they adjust to new contexts. Investigating the concept of home within migratory experiences may allow researchers to investigate phenomena that can facilitate the migrant experience and help promote resilience [6]. In the present study, the construct of home was understood more specifically, as psychological home and will be using the definition offered by Sigmon and colleagues [7]. This concept looks at the way physical construct of a house/dwelling is changed, modified, and interacted with to create a sense of home. 

Currently, the concept of psychological home is not well studied within the research literature, especially within migration contexts. However, understanding how psychological home is experienced by migrants is key to understanding phenomena that may improve acculturative outcomes for migrants. The present study explored whether migrants and Italian citizens reported significantly different sense of psychological home and also whether similar or different factors predicted Italians’ and migrants’ psychological home. We believe this information permits researchers to determine whether findings on psychological home from general populations might be generalizable to migrant populations. 

### 1.1. Psychological Home

This study predominantly examines the concept of psychological home as described by Sigmon and colleagues [7]. Psychological home is defined by Sigmon and colleagues [7] ( p. 11) as “a sense of belonging in which self-identity is tied to a particular place.” These researchers identified five components that comprise the concept of psychological home; namely, the cognitive, behavioral, affective, manifestation and functional component. The cognitive component deals with the beliefs individuals have about the home concept and how much of their identity is tied to the home concept. The behavioral component deals with the deals with the actions that are taken to change the house to become more homely while the affective will look at the feelings and emotions that are typically associated with the home. According to the authors, these feelings are typically developed in early childhood as individuals interact with their early environments such as their home of origin, they also contend that family and environment will likely be key factors in this component. The final two components include the manifestation component which will include how much an individual invests in construction and maintenance in the home environment and are more likely to be devastated when this is somehow damaged or destroyed (this is supported by Carroll and colleagues [8] who looked at the impact of a flood on residents in Carlisle). Finally, Sigmon and colleagues [7] claimed that there is a functional component dealing with the benefits or liabilities obtained from a relationship with the physical home space. 

Psychological home has been consistently associated with life satisfaction in both Italy [9] and America [2,3,10]. For instance, using data gathered in Italy, psychological home predicted resilience in both migrant and non-migrant populations [5]. In the original validation study for this scale Sigmon and colleagues [7] also found psychological home correlated with the Mental Health Index [11] and more specifically was positively correlated with the psychological wellbeing, affiliation, hope and desirability of control subscales [7]. They also found that psychological home was negatively correlated with psychological distress, state-trait anxiety and negative [7]. While most of the studies present evidence of the positive effects of psychological home, Roster and colleagues [3] noted the dark side of psychological home when the dwelling is burdened by physical clutter and reduce the life satisfaction benefits that are provided by psychological home.

Various factors impact psychological home. In their study carried out with Italian adult citizens Cicognani [9] looked at several group differences namely gender, age, education, employment, type of housing, and relationship to housing. For instance, female participants scored significantly higher than male participants on a sense of home. A small correlation was observed between age and psychological home, with mean scores for psychological home increasing with age. These findings were replicated with migrants by Cuba and Hummon [12] and Boccagni and colleagues [13], who both found (in the United States and, Italy, Spain and England, respectively) younger migrants identify a sense of home as pertaining more to relationships. Conversely older migrants especially those who spend more time in the new community equated home with the physical abode [12,13]. 

However, Cicognani’s study [9] also found other demographic factors related to psychological home and these have yet to be studied within the context of migrants. Participants with high school degrees and compulsory schooling scored statistically significantly higher than those with a university degree [9]. A significant difference in psychological home scores was also observed when taking into account the type of house or residence with persons in detached houses scoring the highest and persons living in small apartments scoring the lowest [9]. Finally, significant differences were also observed in the housing situation with owners having higher mean scores than participants who rented [9]. Different employment statuses were not found to have a statistically significant different mean score in psychological home in the Cicognani study [9].

### 1.2. Psychological Home in the Migration Context

Within migration contexts, psychological home plays an important role throughout the acculturation process. Since migrants experience great acculturative stresses when leaving a context of greater familiarity to move into one with less, [14] any concept which may mitigate such stress becomes highly important for researchers. Three studies predominantly look at psychological home in migrants. The first is a scoping review looking at what the literature in a variety of fields has said on migrants and home [15], the second is a qualitative study looking at conceptions of psychological home in male migrants [16] and the third is a study looking at the relationship between psychological home, sense of community and neighborhood attachment, and resilience in migrants [6]. 

Romoli and colleagues [15] identify three aspects emerging from the literature relating to psychological home, much in line with Sigmon and colleagues [7] conceptualizations. The first is the cognitive component wherein conceptualizations about the home and domestic duties were seen to be heavily influenced by issues of competing and a multiplicity of identities [15]. These included factors such as gender [17,18] and sharing spaces with same language speakers [19]. The second component, the affective, relates to the emotional connection migrants feel to their new living spaces [15]. Here, Boccagni [20] specifies that inadequate housing conditions might lead to a lesser attachment to the home spaces, while Cain and colleagues [21] indicate that home ownership also affects psychological home as it adds a sense of permanence to the migration experience (see also [22]). Eskela [23] however finds that homeownership might be a mere economic choice and not necessarily representative of an emotional attachment. The third component which affects migrants’ psychological home is the environment, with stark differences in domestic environment between the home of origin and the new country possibly detracting from sense of psychological home, while investment in relics, symbols and other objects of the country of origin being conducive to greater psychological home [15].

In another qualitative study Cardinali and colleagues [24] found that a sense of psychological home by male migrants represented a base of safety where self-identity could develop, still within the primary control of women, and was highly influenced by long-term employment prospects. Camilleri [6] also found that employment positively predicted psychological home in migrants and that in turn psychological home predicted resilience in both migrants and Italians. Interestingly within this study, sense of community and neighborhood attachment did not predict resilience within both migrants and Italians, thereby highlighting the salience of psychological home, particularly for migration research.

Taken together, there appears to be a scarcity of studies relating to the psychological home construct, particularly when assessing the link between varied socio-demographic variables and psychological home. Identifying consistent predictive relationships between socio-demographic variables and psychological home is imperative for researchers within the field of community psychology to understand the contextual factors that influence the psychological home construct. Additionally, given the scarcity of research on psychological home and it’s important to migrant populations it is important to ascertain whether migration status changes the predictive relationship between socio-demographic variables and psychological home. Presently, while a small number of studies focused on psychological home have yielded consistent results across multiple sites, barring the three studies mentioned, migrants’ sense of psychological home is severely understudied. Addressing this gap in current knowledge will help policy makers and community organizations organize evidenced-based programming for migrants. Expanding our knowledge on psychological home between general populations and migrants will contribute to narrowing the gap in the literature.

The present study explored whether migrants and Italian citizens reported significantly different sense of their psychological home scores and whether different self-reported demographic variables affect psychological home across these groups. We believe this information will permit researchers to further illustrate the relationship between migrants and citizens with regard to psychological home. I. It will do so by assessing whether migration status (i.e., Italian citizenship and migrants) influences the relationship between each of the following demographic variables: age, gender, education, employment, type of housing lived in, relationship to housing, and psychological home. 

Given that no previous literature exists, this research will provide a foundation for future research on the matter, despite not being representative of both citizen and migrant populations within Italy. Additionally, the novel nature of the research will inform the prediction of the outcome of this analysis, utilizing foundational principles of community psychology to inform the directions of hypothesis. Such principles acknowledge the imperialist, colonial, and racist practices that cross-cultural research (which this study participates in by comparing two highly different groups) has typically engaged in [24]. Informed by a desire to avoid such practices, differences will for the most part be assumed within the migrants and citizens to allow for conceptual space for variation given the highly different life experiences of migrants and citizens.

### 1.3. Hypotheses

The present study had **seven** hypotheses, following Cicognani’s [9] findings. 

**Hypothesis** **I.**
*There is a significant difference in psychological home mean scores between genders, and there is a significant difference between migrants and Italians psychological home scores. The effect of gender will depend on the effect of migration status.*


**Hypothesis** **II.***There is a significant difference in psychological home mean scores between different levels of education, and there is a significant difference between migrants and Italians psychological home mean scores. The effect of education will depend on the effect of migration status*.

**Hypothesis** **III.***There is no significant difference in psychological home mean scores according to occupation status, and there is no significant difference between migrants and Italians psychological home mean scores. The effect of occupations status will not depend on the effect of migration status*.

**Hypothesis** **IV.***There is a significant difference in psychological home mean scores between different types of housing, and there is a significant difference between migrants and Italians psychological home mean scores. The effect of different types of housing relationships will depend on the effect of migration status*.

**Hypothesis** **V.**
*There is a significant difference in psychological home mean scores in different housing relationships, and there is a significant difference between migrants and Italians psychological home mean scores. The effect of different types of housing relationships will depend on the effect of migration status.*


**Hypothesis** **VI.***Age will correlate significantly with psychological home mean scores and there will be a significant difference between the correlation coefficient of migrants and Italians*.

## 2. Methods

### 2.1. Procedure

This study utilized archival on-line survey data collected by researchers from the University of Genoa in Italy. The study was conducted in accordance with the Declaration of Helsinki, and approved by Ethics Committee of Dipartimento di Scienze della Formazione (DISFOR) (CER_035_1(002). Data gathering took place between May and December 2020 through survio.com. The survey took around 20 minutes to complete and participants (*n* = 208) were required to complete a consent. Migrant participants were recruited through contacts researchers had with embassies and migrant NGOs in Italy, while a snowball sampling technique was used to recruit Italian citizens. Persons indicated either they lived in Italy as residents or migrated there. Participants were required to be 18 years or older to participate in the survey. Their age ranged from 18 to 80 years (*M* = 35.16, *SD* = 13.63). The survey was presented in Italian and all participants understood Italian.

### 2.2. Participants

Participants were divided into two groups, Italians (*n* = 76) or migrants (*n* = 132). The migrant population consisted of 82 women and 50 men (*M* age = 34.38 years, *SD* = 12.78). Most migrants originally migrated from Albania *(n* = 53), Romania (*n* = 23), or Ecuador (*n* = 14). Additionally, most migrant respondents worked full time (*n* = 45), claimed a high school diploma (*n* = 37), lived in an apartment (*n* = 116) or owned their house (*n* = 66). Table 1 presents detailed demographic information on migrants.

Italians comprised of 52 women and 24 men (*M* age = 36.63, *SD* = 14.97). Most Italians were employed full-time (*n* = 40), reported a high school diploma (*n* = 28), and lived in an apartment (*n* = 54) or owned their housing (*n* = 54). Table 1 also contains information on the Italian’s demographic information. A chi-square analysis and a t-test of the various demographics was conducted between migrants and citizens, with employment (*p* < 0.001), type of housing lived in (*p* = 0.002), relationship to housing being statistically (*p* = 0.017) significantly different. Citizens tended to have more full-time employment, tended to live in single-family houses, and were more likely to own their homes. Given the disparity in resources and opportunities to accrue wealth and employment opportunities [5] with their subsequent effect on housing, these differences were deemed to be inherent to the nature of migratory populations. Therefore, unless participants, particularly migrants within particular demographics are over-represented by design, it is likely that most studies comparing migrants and citizens will have statistically significant differences in demographics. 

### 2.3. Table 1—Particiant Profile, by Demographic Variables

**Table 1 behavsci-12-00387-t001:** Percentages by Demographic Variables.

	Migrant (*n* = 132)	Italians (*n* = 76)
	%	%
*Gender*		
Male	37.90	31.60
Female	62.10	68.40
*Employment*		
Full-time worker	34.10	52.60
Part-time worker	24.20	5.30
Unemployed	15.90 **	2.60 **
Student	25.00	31.60
Retired	0.80 **	7.90 **
*Education Status*		
Primary School	2.30	1.30
Secondary School	15.20	6.60
Professional Qualification	12.90	7.90
High School Diploma	28.00	36.80
Three Year Degree	18.20	23.70
Masters	22.00	21.10
Other	1.50 *	2.60 *
*Dwelling Type*		
Apartment	87.90	71.10
Single-Family House	8.30	19.70
Multi-Family House	80.00	9.20
Studio Apartment	1.50 *	0.00 *
University Residence	1.50 *	0.00 *
*Home Ownership*		
Owner	50.00	71.10
Renter	48.50	28.90
Living free of charge	1.50 *	0.00 *

*Note. N* = 208 (132 migrants, 76 Italians). * = Removed because of small cell size, ** = Removed because the unequal cell sizes affected the homogeneity of variance.

### 2.4. Psychometric Measures

Italian Psychological Home was measured using the 8-item *Psychological Home Scale* initially developed by Sigmon and colleagues [6] and translated in Italian by Cicognani [9]. According to Sigmon and colleagues [6], the Psychological Home Scale measured the relationship that one has to their physical dwelling space. Item responses ranged from 1 (strongly disagree) to 7 (strongly agree). Sample items include “I get a sense of security from having a place of my own.” and “I surround myself with things that highlight my personality”. (*M* = 5.52, *SD* = 1.10, α = 0.820)

### 2.5. Statistical Analysis

Five two-way Anovas were carried out after checking for assumptions. The two-way Anovas were utilized to assess for significant difference between Italians and migrants when factoring gender, education, employment, types of housing, and housing relationship with respect to their psychological home scores. A further analysis was carried out using a Pearson product-moment correlation to determine the relationship between age and psychological home in both migrants and Italians separately. Once the two r-scores were determined the r-scores were compared using an online calculator which ran a Fisher’s R to Z transformation [25]. 

## 3. Results

### 3.1. There Is a Significant Difference in Psychological Home Mean Scores between Genders, and There Is a Significant Difference between Migrants and Italians Psychological Home Scores. The Effect of Gender Will Depend on the Effect of Migration Status

A *two-way ANOVA* examined the effects of immigration status and gender on psychological home. Data reported are mean ± standard deviation, unless otherwise stated. Residual analysis was performed to test for the assumptions of the two-way ANOVA. Outliers were assessed by inspection of a boxplot; homogeneity of variances was assessed by Levene’s test. There were four outliers that were removed, and there was homogeneity of variances (*p* = 0.818).

There was a statistically significant interaction between gender and immigration status on psychological home, *F* (1, 200) = 4.394, *p* = 0.037, partial η^2^ = 0.021. There was a statistically significant difference in psychological home for gender (*p* = 0.002) but not for migration (*p* = 0.109) Therefore, an analysis of simple main effects for migration status was performed. There was a statistically significant difference in mean Psychological home scores for Italians males and females, *F* (1, 200) = 10.46, *p* < 0.0001, partial η^2^ = 0.050, but not for migrants, *p* = 0.387.

All pairwise comparisons were run for each simple main effect with reported 95% confidence intervals. Mean Psychological home scores for migrant and Italian females were 5.69 (*SD* = 0.93), 4.97 (*SD* = 1.13), respectively. Migrant females had a statistically insignificant higher “Psychological home” score than Italian females *p* = 0.677.

Mean “Psychological Home” scores for migrant and Italian males were 5.53 (*SD* = 1.08) and 4.97 (*SD* = 1.13), respectively. Migrant males had a statistically significantly higher mean “Psychological home” score than Italian males, mean difference = 0.56, 95% CI [0.076, 1.047], *p* = 0.024 as can be seen in Table 2. 

The hypothesis was partially supported with significant differences being found between Italian males and females, and significant differences being found between Italian and migrant males. 

### 3.2. Hypothesis II. There Is a Significant Difference in Psychological Home Mean Scores between Different Levels of Education, and There Is a Significant Difference between Migrants and Italians Psychological Home Mean Scores.The Effect of Education Will Depend on the Effect of Migration Status

A *two-way ANOVA* examined the effects of immigration and educational level on psychological home. Residual analysis was performed to test for the assumptions of the two-way ANOVA. Outliers were assessed by inspection of a boxplot and homogeneity of variances was assessed by Levene’s test. There were three outliers that were removed, and there was homogeneity of variances (*p* = 0.249).

The interaction effect between immigration status and educational level on psychological home was not statistically significant, *F* (6, 191) = 2.161, *p* = 0.056, partial η^2^ = 0.062. There was no statistically significant difference in “Psychological home” score for immigration status and education *p* = 0.530 and 0.753, respectively. Therefore, the hypothesis was not supported. 

### 3.3. Hypothesis III. There is no significant difference in psychological home mean scores according to occupation status, and there is no significant difference between migrants and Italians psychological home mean scores. The effect of occupations status will not depend on the effect of migration status

Another *two-way ANOVA* explored the effects of immigration and employment status on psychological home. Residual analysis was performed to test for the assumptions of the two-way ANOVA. Outliers were assessed by inspection of a boxplot and homogeneity of variances was assessed by Levene’s test. There were three outliers that were removed, and there was no homogeneity of variances (*p* = 0.039). To address the heterogeneity of variances, the unemployed and retired categories were removed given the very small number of observations present within these cells (n = 23 and n = 7, respectively). After this correction the assumption of homogeneity of variances was satisfied (*p* = 0.115).

The interaction effect between immigration status and employment status on psychological home was not statistically significant, *F* (2, 169) = 1.02, *p* = 0.365, partial η^2^ = 0.012. There was no statistically significant difference in psychological home score between immigrants and Italians, *F*(1, 169) = 0.613, *p* = 0.435, partial η^2^ = 0.004. A statistically significant difference was found in psychological home score for the different employment statuses *F*(2, 169) = 3.95, *p* = 0.021, η^2^ = 0.045.

Pairwise comparisons were run where reported with 95% confidence intervals and p-values were Bonferroni-adjusted. The unweighted marginal means of psychological home scores for persons employed full-time, part-time, and students were 5.82 ± 0.106, 5.54 ± 0.258 and 5.35 ± 0.132, respectively.

Full-time employment status was associated with a mean psychological home scores 0.474 (95% CI, 0.063 to 0.884) points higher than a student employment status, a statistically significant difference, *p* < 0.018. Full-time employment status did not have a significant different in mean psychological home scores from part-time employment status *p* = 0.936 and neither did part-time employment status and student *p* = 1.00.

### 3.4. Hypothesis IV. There Is a Significant Difference in Psychological Home Mean Scores between Different Types of Housing, and There Is a Significant Difference between Migrants and Italians Psychological Home Mean Scores.The Effect of Different Types of Housing Relationships Will Depend on the Effect of Migration Status

A *two-way ANOVA* examined the effects of immigration and types of housing on psychological home. Residual analysis was performed to test for the assumptions of the two-way ANOVA. Outliers were assessed by inspection of a boxplot and homogeneity of variances was assessed by Levene’s test. There were four outliers that were removed, and there was homogeneity of variances (*p* = 0.268).

The interaction effect between immigration status and type of dwelling on psychological home was not statistically significant, *F*(2, 196) = 0.319, *p* = 0.727, partial η^2^ = 0.003. There was no statistically significant difference in “Psychological home” score for immigration status and type of dwelling *p =* 0.370 and 0.987, respectively. Therefore, the fourth hypothesis was not supported. 

### 3.5. Hypothesis V. There Is a Significant Difference in Psychological Home Mean Scores in Different Housing Relationships, and There Is a Significant Difference between Migrants and Italians Psychological Home Mean Scores.The Effect of Different Types of Housing Relationships Will Depend on the Effect of Migration Status

A *two-way ANOVA* then examined the effects of immigration and different housing relationships on psychological home. Residual analysis was performed to test for the assumptions of the two-way ANOVA. Outliers were assessed by inspection of a boxplot and homogeneity of variances was assessed by Levene’s test. There were four outliers that were removed, and there was homogeneity of variances (*p* = 0.064).

The interaction effect between immigration status and type of dwelling on psychological home was not statistically significant, *F* (1, 201) = 0.731, *p* = 0.394, partial η^2^ = 0.004. There was no statistically significant difference in “Psychological home” score for immigration status and relationship to dwelling *p* = 0.415 and 0.304, respectively. Therefore, the fifth hypothesis also was not supported.

### 3.6. Hypothesis VI. Age Will Correlate Significantly to Psychological Home Mean Scores and There Will Be a Significant Difference between the Correlation Coefficient of Migrants and Italians

A Pearson product-moment correlation was run to determine the relationship between age and psychological home in both migrants and Italians. For migrants four outliers were removed and there was a small, positive correlation between age and psychological home scores, which was statistically significant (*r* = 0.235, *n* = 128, *p* = 0.008) with age accounting for 5% of variance for mean scores of psychological home in migrants. 

For Italians seven outliers were removed and there was a moderate, positive correlation between age and psychological home scores, which was statistically significant (*r* = 0.316, *n* = 0.69, *p* = 0.008) with age accounting for 9% of psychological home in Italians.

The strength of the two r scores, for migrants and Italians (0.235 and 0.316, respectively), were compared using an online calculator which ran a Fisher’s R to Z transformation [25]. The two r values were not found to be significantly different *p* = 0.284. This hypothesis, therefore, was only partially supported with significant differences being found in age for migrants and Italians, respectively, but these were not found to be significantly different between them.

## 4. Discussion

Research on psychological home has given consistent results in its associative and predictive power for wellbeing and life satisfaction, thereby rendering the construct of great importance to underserved populations whose housing spaces might be precarious and, in some instances, non-existent. Consequently, psychological home becomes a highly important construct within migration experiences ostensibly providing a buffer against acculturative stresses and possibly discriminatory regimes.

Previous literature [9] on psychological home within Italy highlighted significant differences in psychological home scores for a number of cohorts including: females scoring more highly than males, lower education levels having significantly higher scores than university level participants, and no significant differences on scores based on employment. Additionally, significant differences were found between different types of housing (with detached houses scoring higher than apartments), between owners and renters (with the latter having higher psychological home scores), and finally a small correlation between age and psychological home (with higher psychological home for older participants). However, the findings of the Cicognani study [9] have not yet been researched within migration contexts. Given this paucity of literature the present study explored whether (1) previous findings for non-migrant populations were replicated, and (2) whether such findings generalized to migrant populations. 

Results of the present study indicated overall there are no significant differences between migrants and Italians for psychological home scores, with significant interactions only being observed in the case of gender (H. I). With regard to education (H. II), employment (H. III), types of housing (H. IV), and relationship to housing (H. V) no significant interaction was observed both between migrants and Italians and within the categories themselves. Employment was statistically as a main effect with significant differences in psychological home scores observed between full-time employees and students. With regard to age (H. VI), significant differences were not observed between migrants and Italians; however, within each sample age correlated with psychological home. Within migrants a low positive correlation was observed while for Italians a moderate positive correlation was observed.

The findings of the present study contrasted with some of Cicognani’s [8] findings with type of housing, relationship to housing and education not having statistically significant different levels of psychological home, both for Italians and for migrants. Additionally in contrast to the Cicognani study [9], employment was found to have statistically significant differences between various employment statuses but not between migrants and Italian citizens. The other findings of the present study confirm findings of the Cicogani [9] study with age psychological home but did not vary significantly between migrants and gender having statistically significant differences in psychological home scores and a statistically significant difference between migrants and Italian citizens. A potential reason for the discrepancy within these findings is that data collection occurred during a historic event namely the COVID-19 pandemic. Whereas in non-pandemic times differences in psychological home scores are easier to differentiate when looking at various socio-demographic factors, lockdowns and prolonged time spent at home may have dampened some of these key differences. 

The findings of the present study strongly support the notion that the psychological home construct does not differ significantly between migrant and non-migrant populations. In addition, the effects of education, employment, type of housing, relationship to housing and age on psychological home do not appear to differ between migrants and Italians. These findings therefore support the idea that researchers may generalize their findings from the general population to migrant populations, with one important exception, gender.

Within the six factors looked at within this study, gender was the only construct that affected the relationship between citizenship and psychological home. The findings in this study show that while Italians had statistically significant differences between male and female psychological home scores, this was not found to the case for migrants. Additionally, when comparing the different genders according to immigration status, migrant and Italian citizen females did not differ significantly in their psychological home scores while males did. 

An explanation for these two phenomena might reside in the fact the migration experience might heighten the sense of psychological home for both males and females and reduce the impact of internalized gender roles. Given the acculturative stresses migrants experience, male migrants in particular might identify more greatly with psychological home as this would be considered the primary safe space rather than the broader community [23]. In effect migration status might account for a reversal of gender norms when relating to the psychological home. For non-migrant populations however, traditional gender roles might prevail in key differences in sense of psychological home. 

Such an explanation would also account for the reason why male migrants have a statistically significant higher psychological home score then Italian males. Italian males might relate to the home construct in ways which are more indicative of traditional gender norms and therefore identify more closely with external constructs such as community membership. This would render the identification with the home construct weaker, particularly when compared to migrant males who might not have the same opportunities for external socialization. 

The present study had several limitations in relation to the size and composition of the sample. While overall the sample size was of a good level (*n* = 208), the unequal migrant and Italian sizes might limit some of the findings in this study. Additionally, the data utilized within this study was gathered throughout the COVID-19 pandemic which required significant alteration to daily routines and may have influenced the relationship individuals had with their home given that they were spending significant amount of time in it. A third limitation was that the composition of the migrant population is not statistically representative of migrant demographics within Italy. Therefore, some limitations as to generalizability also exist. Nonetheless, the overall finding corroborated through the various analyses carried out in this study still indicate that overall, migrant and non-migrant populations do no not differ significantly on many major factors associated with the psychological home construct. However, further research with more representative samples is required to fully assess this claim.

One key finding that differs significantly from literature on non-migrant populations is gender differences in psychological home. Future studies can investigate more deeply whether this moderating relationship diminishes over time as migrant males gain more access to external resources such as community organizations or whether this moderation holds true despite the length of stay within the host community. The dimension of temporality and its impacts on demographics and psychological home will also be an important avenue for further investigation and which have not been investigated in this study.

The implication of this present study for policy makers, community organizations and community psychology is that future programming for migrants can continue by and large to use the findings of mainstream research to inform its contents. Additionally, whereas in previous research [9] significant differences were found for varying education levels, and different housing situations, this was not found to be the case in the present study. Employment held significant differences in psychological home scores would therefore corroborate Camilleri’s [6] finding. This would suggest that resources spent on employment are more likely to heighten psychological home with its subsequent benefits than more education, larger homes, and home ownership. 

Future research can build on the present study by investigating whether the effect of migration and gender persists over time or whether further acculturation and presumably more access to communal resources will result in citizen males and migrant males having similar psychological home scores. Additionally, more nuanced methods to ascertain migrants and Italians understanding of home as well as to measure employment including salary levels may be used in future studies to understand further the relationship between migration status and psychological home. 

In summary, our research confirmed the importance of gender and more limitedly age as important constructs across migrant and non-migrant populations within a sense of psychological home. Despite the lack of representativeness of our sample, this study contributed to the literature by providing a foundational understanding on psychological home for non-migrant populations and the similarities to migrant populations. 

## Figures and Tables

**Table 2 behavsci-12-00387-t002:** 2-way Anova Gender, Education, Occupation, Type of Housing, Housing Relationship, and Immigration Status.

Variable		Sum of Squares	*df*	Mean Square	F	*p*	Partial η^2^
Gender							
	Intercept	5113.34	1	5113.34	5235.54	0.000	0.963
	Gender	9.51	1	9.51	9.74	0.002	0.046
	Immigration Status	2.53	1	2.53	2.59	0.109	0.013
	Gender*Immigration Status	4.29	1	4.29	4.39	0.037	0.021
	Error	195.33	200	0.98			
Education							
	Intercept	1841.39	1	1841.39	1780.20	0.000	0.903
	Education	3.54	6	0.59	0.57	0.753	0.018
	Immigration Status	0.41	1	0.41	0.40	0.530	0.002
	Education*Immigration Status	12.97	6	2.16	2.09	0.056	0.062
	Error	197.56	191	1.03			
Occupation							
	Intercept	2767.25	1	2767.25	2922.75	0.000	0.945
	Occupation	0.58	1	0.58	0.61	0.021	0.004
	Immigration Status	7.49	2	3.74	3.95	0.435	0.045
	Occupation*Immigration Status	1.92	2	0.96	1.01	0.365	0.012
	Error	160.01	169	0.95			
Type of Housing							
	Intercept	894.12	1	894.12	853.88	0.000	0.813
	Type of Housing	0.35	4	0.09	0.08	0.987	0.002
	Immigration Status	0.84	1	0.84	0.81	0.370	0.004
	Type of Housing*Immigration Status	0.67	2	0.33	0.32	0.727	0.003
	Error	205.24	196	1.05			
Housing Relationship							
	Intercept	881.28	1	881.28	792.36	0.000	0.798
	Type of Housing	2.66	2	1.33	1.20	0.304	0.012
	Immigration Status	0.74	1	0.74	0.67	0.415	0.003
	Type of Housing*Immigration Status	0.81	1	0.81	0.73	0.394	0.004
	Error	223.55	201	1.11			

*Note. n* = 208 (132 migrants, 76 Italians).

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
