# Peer review of "Home Away from Home: Comparing Factors Impacting Migrants’ and Italians Sense of Psychological Home"

_behavsci, 2022, doi:10.3390/bs12100387_

Round 1

Reviewer 1 Report

This "little" study, small number of participants, particularly Italians, is important and should be able to generate some follow-up dialogue among scholars in the field.  The paper starts with very strong literature review, and the study was informed by strong theoretical framework.  The findings are intriguing that most of the variables are not statistically significant among each other.  This begs for hypothetical interpretation of the results.

The authors also highlighted the covid experience might skew the results in the findings.  Very possible, but this study can be easily replicated at other times that might eliminate the "covid effect".

However, as its limitations stated clearly in the conclusion, this type of study needs larger sample size which should not be difficult to find or collect.  Furthermore, some follow-up in-depth interviews from the sample, if possible at all, would be fruitful to discover what home really means to both migrants and non-migrants under the variables tested here.  This can be highlighted in the discussion and conclusion.

Otherwise, this paper meets the expectations in its own rights.

Author Response

Reviewer 1

This "little" study, small number of participants, particularly Italians, is important and should be able to generate some follow-up dialogue among scholars in the field.  The paper starts with very strong literature review, and the study was informed by strong theoretical framework.  The findings are intriguing that most of the variables are not statistically significant among each other.  This begs for hypothetical interpretation of the results.

The authors also highlighted the covid experience might skew the results in the findings.  Very possible, but this study can be easily replicated at other times that might eliminate the "covid effect".

However, as its limitations stated clearly in the conclusion, this type of study needs larger sample size which should not be difficult to find or collect.  Furthermore, some follow-up in-depth interviews from the sample, if possible at all, would be fruitful to discover what home really means to both migrants and non-migrants under the variables tested here.  This can be highlighted in the discussion and conclusion.

Otherwise, this paper meets the expectations in its own rights.

Dear Reviewer,

Thank you for the insightful feedback. We agree that further in-depth investigation could strengthen the paper.

However, given that the data was collected in May-December 2020 and the migrant population tends to be quite transient, it would be difficult to track the individuals involved in the original study. We have mentioned this suggestion as a future direction in line 488-489.

Regards,

Andrew P. Camilleri

Reviewer 2 Report

The text presents a methodological problem that affects the results.

We do not know the universe of the population. Therefore, the data are not representative. This makes the work more of a case study. A matter that should be taken into account from the title to the whole structure of the work.

Since it seems that the results are statistically significant and can be extended to all migrants.

However, this does not detract from the work, but its orientation and presentation should be somewhat different when it comes to expressing the results.

Therefore, this work should reflect that it is, in itself, a starting point, as a hypothesis, for similar work with Italian migrants.

And this issue should be well reflected in the introduction and final conclusions. 

Author Response

Reviewer 2

The text presents a methodological problem that affects the results.

We do not know the universe of the population. Therefore, the data are not representative. This makes the work more of a case study. A matter that should be taken into account from the title to the whole structure of the work.

Since it seems that the results are statistically significant and can be extended to all migrants.

However, this does not detract from the work, but its orientation and presentation should be somewhat different when it comes to expressing the results.

Therefore, this work should reflect that it is, in itself, a starting point, as a hypothesis, for similar work with Italian migrants.

And this issue should be well reflected in the introduction and final conclusions. 

Dear Reviewer,

Thank you for the insightful feedback.

We acknowledge the limitations of our sample and agree that this work is a foundational study that can provide insights to future work with more representative sample within their work. Acknowledging this factor, we have made changes in lines: 151-153, 159-162, 463-464, 494-496.

Your faithfully,

Andrew P. Camilleri

Reviewer 3 Report

This is an interesting topic of relevance to the study of migration and its impact. The examination of psychological home" is important to understanding of the overall wellbeing of not only migrants but of all people. 

The authors have provided a good context and conceptualization for the proposed study and have done a nice job of situating the research questions on the research literature.

Here are some questions, responses to which would make the manuscript clearer.

Is it safe to assume that all the migrants spoke and understood Italian; and they responded to the survey in Italian? 

What is the rationale for the selected analytical method used and if possible, what other analytical methods could have been tried (were it not for limitations of the data, etc. as the authors discuss  in the paper).

Do the authors have any data on how long the migrants have been in Italy?  What impact does length of residence in Italy may factor in "psychological home" ?  If  this is a limitation of the study, it should be mentioned. 

Were there any statistically significant differences between migrants and the Italians on demographic factors? If so, what are the implications for interpreting the results? 

Author Response

This is an interesting topic of relevance to the study of migration and its impact. The examination of psychological home" is important to understanding of the overall wellbeing of not only migrants but of all people. 

The authors have provided a good context and conceptualization for the proposed study and have done a nice job of situating the research questions on the research literature.

Here are some questions, responses to which would make the manuscript clearer.

Is it safe to assume that all the migrants spoke and understood Italian; and they responded to the survey in Italian? 

  • Dear Reviewer, thank you for the insightful feedback. Yes, all migrants spoke and understood Italian in the survey. We have added a sentence to clarify this in line 215.

What is the rationale for the selected analytical method used and if possible, what other analytical methods could have been tried (were it not for limitations of the data, etc. as the authors discuss in the paper).

  • Dear Reviewer, thank you for the insightful feedback. Given the categorical nature of the data in Hypothesis 1-5, a 2-way ANOVA was deemed as best suited to identify significant differences across demographic variables and migrant status. With regards to Hypothesis 6, since the data was continuous a Pearson’s correlation which was then subjected to Fisher’s R-to-Z transformation was deemed best at identifying whether significant differences existed between migrants and citizens with regards to age and psychological home.

Do the authors have any data on how long the migrants have been in Italy?  What impact does length of residence in Italy may factor in "psychological home" ?  If  this is a limitation of the study, it should be mentioned. 

  • Dear Reviewer, thank you for the insightful feedback. Length of stay is a limitation of this study and we have acknowledged this further in line 471-474

Were there any statistically significant differences between migrants and the Italians on demographic factors? If so, what are the implications for interpreting the results? 

  • Dear Reviewer, thank you for the insightful feedback. We ran additional analyses for demographic factor differences between migrants and citizens and provided an interpretation on lines 226-236

Yours faithfully,

Andrew P. Camilleri

Round 2

Reviewer 2 Report

The text has been improved

Author Response

Thank you for your comments.